# Prolonged Differentiation of Neuron-Astrocyte Co-Cultures Results in Emergence of Dopaminergic Neurons

**DOI:** 10.3390/ijms24043608

**Published:** 2023-02-10

**Authors:** Victoria C. de Leeuw, Conny T. M. van Oostrom, Edwin P. Zwart, Harm J. Heusinkveld, Ellen V. S. Hessel

**Affiliations:** Centre for Health Protection (GZB), National Institute for Public Health and the Environment (RIVM), Antonie van Leeuwenhoeklaan 9, 3721 MA Bilthoven, The Netherlands

**Keywords:** neurotoxicity, neurodegeneration, human embryonic stem cells, dopaminergic neurons, in vitro

## Abstract

Dopamine is present in a subgroup of neurons that are vital for normal brain functioning. Disruption of the dopaminergic system, e.g., by chemical compounds, contributes to the development of Parkinson’s disease and potentially some neurodevelopmental disorders. Current test guidelines for chemical safety assessment do not include specific endpoints for dopamine disruption. Therefore, there is a need for the human-relevant assessment of (developmental) neurotoxicity related to dopamine disruption. The aim of this study was to determine the biological domain related to dopaminergic neurons of a human stem cell-based in vitro test, the human neural progenitor test (hNPT). Neural progenitor cells were differentiated in a neuron-astrocyte co-culture for 70 days, and dopamine-related gene and protein expression was investigated. Expression of genes specific for dopaminergic differentiation and functioning, such as *LMX1B*, *NURR1*, *TH*, *SLC6A3*, and *KCNJ6*, were increasing by day 14. From day 42, a network of neurons expressing the catecholamine marker TH and the dopaminergic markers VMAT2 and DAT was present. These results confirm stable gene and protein expression of dopaminergic markers in hNPT. Further characterization and chemical testing are needed to investigate if the model might be relevant in a testing strategy to test the neurotoxicity of the dopaminergic system.

## 1. Introduction

Dopamine, one of the neurotransmitters of the central nervous system, plays a crucial role in a series of biological processes during brain development and throughout life. The cells that produce dopamine are predominantly located in the central parts of the brain and are vital for executive and motor function, motivation, cognition, and reward. Dopamine is already present in early brain development even prior to synaptogenesis, and activation of dopamine receptors during development alters brain structure and connectivity with enduring anatomical and behavioral effects throughout adulthood [1,2]. 

Disruption of the dopaminergic system may result in a variety of brain disorders [1,3,4,5]. The best-known example is Parkinson’s disease (PD). PD is characterized by the degeneration of dopaminergic neurons in the substantia nigra pars compacta (SNpc), disrupting the nigrostriatal signaling pathway [6]. Clinical features of PD are typically seen when 60–70% of dopaminergic neurons in the SNpc are lost [7,8]. The causes of PD are multifactorial and involve genetic, lifestyle, and environmental factors [1,9,10]. Among the environmental factors, pesticides are among the most mentioned to play a role in the etiology of PD [11], and there has been an increasing number of epidemiological studies underlining the association between pesticide exposure and PD [9,12,13,14,15,16]. Furthermore, dopaminergic signaling is suggested to play a role in neurodevelopmental disorders, such as schizophrenia, attention deficit hyperactivity disorder (ADHD), or autism spectrum disorders [1]. However, the exact roles of dopaminergic signaling in these diseases are far from fully understood. Recently, a hypothesis has been published concerning how chemical exposure contributes to the etiology of behavioral problems such as ADHD [17].

The link between chemical exposure, disruption of the dopaminergic system, and (developmental) neurotoxicity ((D)NT) in humans is challenging to prove given the complexity of the brain, the small cohorts of patients, and complex exposure profiles during the life course. Therefore, there is a need for a more human-relevant assessment of (D)NT using a combination of relevant in vitro assays based on human physiology [18,19]. In vitro testing batteries should be based on mechanistic knowledge from human physiology and disease and structured in adverse outcome pathways (AOPs). Given the role of dopamine in brain function, dopaminergic signaling is a crucial element in a testing strategy for (D)NT. Human stem cell-based in vitro assays can be of great use to study the potential effects of compounds on (developing) dopaminergic neurons. There is great progression in the differentiation of human stem cells towards a dopaminergic fate for therapeutical purposes (e.g., [20,21]). However, cell cultures, especially co-cultures, suited for the assessment of chemical exposure specifically to dopaminergic neurons are not yet well-developed, but urgently needed [22,23,24].

In our laboratory, we have a human embryonic stem cell (hESC)-derived neuronal differentiation assay for the assessment of compound-induced developmental neurotoxicity: the human neural progenitor test (hNPT) [25]. This multicellular assay consists of glutamatergic neurons, GABA-ergic neurons, and astrocytes, forming a functional neuronal network with spontaneous electrical activity [26]. In this manuscript, we show that prolonged culture of these cells gives rise to dopaminergic neurons. Therefore, this model could be relevant as a complex cell model to study the effect of chemicals on dopamine neurons. It is essential to define the applicability domain of a model for proper placement of this in vitro model in a testing strategy. With this study, we aim to determine the biological domain of the hNPT related to dopaminergic system.

## 2. Results

### 2.1. Robust Gene Expression of Dopaminergic Markers after Four Weeks of Differentiation

Dopaminergic neurons express a set of genes specific for this neuronal population. The first aim was, therefore, to identify if and when dopaminergic markers were expressed over the course of differentiation of the neuron-astrocyte co-culture. The culture period was prolonged to 70 days of differentiation, and samples were taken at different time points (Figure 1A). After 14 days, an increase of most dopamine-related gene markers could be observed, which continued towards robust expression over the weeks after (Appendix A). The order of appearance was more or less in line with data from in vivo development [27,28]. A marker for differentiation of dopaminergic progenitor cells, *LMX1B*, was upregulated in the first 28 days of differentiation and decreased afterward (Figure 1B). Expression of *NURR1*, another transcription factor essential for dopaminergic differentiation, peaked later at 42 days of differentiation, consistent with their sequential expression in development. *PITX3* gene expression, however, showed a large variability without a clear trend. Catecholaminergic neuron marker *TH* reached stable expression from 42 days of differentiation. Two markers for functional dopaminergic neurons, transporter gene *SLC18A2* and potassium channel gene *KCNJ6*, were upregulated similarly to *TH*. Surprisingly, *SLC6A3* gene expression was downregulated from the moment of differentiation.

In addition to the specific markers for dopaminergic cells, markers for general neuronal and astroglial differentiation were assessed. Consistent with previous experiments, markers for ectodermal differentiation (*NEUROG1*, *NES*) were downregulated while markers for neuronal differentiation (*TUBB3*, *MAP2*, *DLG4*, *SYNPR*, *SLC17A6*, *SLC32A1*) and astroglial differentiation (*GFAP*) were upregulated, with most of them stabilizing around 28 days into differentiation (Figure 1C). In short, these results suggest that the neuron-astrocyte co-culture may also contain dopaminergic neurons after prolonged differentiation.

### 2.2. Generating Dopaminergic Neurons after Six Weeks of Differentiation

To confirm the presence of dopaminergic neurons, a series of immunostainings was performed. Images taken at 14, 21, 28, 42, 56, and 70 days of differentiation showed a dense network of neurons that became more segmented into clusters of cells over time, connected by large bundles of neurites (Figure 2A,B). Immunostaining of TH^+^ cells revealed that this group of cells was already present from 14 days of differentiation, increasing in number over time (Appendix A). Cells expressed varying levels of TH in the soma, axons, and dendrites (Figure 2C).

Further characterization of the TH^+^ cells showed that VMAT2 (encoded by *SLC18A2*) was expressed abundantly from 28 days of differentiation (Figure 3A–D, gray and green channel). Expression of DAT (encoded by *SLC6A3*) was present from 42 days onwards, while gene expression of this marker was downregulated relative to the neural progenitor cells (NPCs) ((Figure 3A–D, red channel; Figure 1B). Another interesting observation was that VMAT2 was highly expressed in GFAP^+^ astrocytes (Figure 3E). Together, these results confirm that there was stable protein expression of dopaminergic markers after six weeks of differentiation.

## 3. Discussion

There is a need for human-relevant models that can be used to study how compound exposure is related to brain disorders. Given the complex nature of the brain, a battery of in silico and in vitro is needed. In combination, these models need to cover all potential biological pathways through which a compound may cause brain-related disorders. These biological pathways can be organized in AOPs. AOPs consist of so-called key events, which are measurable steps along the AOP from the point that a compound interacts with a molecular target to the adverse outcome [29]. Once the key events in an AOP are defined, assays can be linked to cover one or more of these key events. Essential for the proper coupling of an in vitro model to a particular key event is a definition of its applicability domain, i.e., which biological processes and underlying mechanisms are and are not represented in this model. The required complexity of the model is, therefore, strongly dependent on its place in an AOP.

AOPs that include the dopaminergic system have been developed [30] for Parkinson motor deficits or neuronal dysfunction. However, in vitro models that realistically represent relevant parts of the human dopaminergic system, allowing for the assessment of key events, such as mitochondrial dysfunction and neuronal degeneration of dopaminergic neurons, are scarce [22,23,24]. Since the dopaminergic system plays a key role in normal brain functioning, models closely resembling the human dopaminergic system are crucial in testing strategies for (developmental) neurotoxicity.

The limited number of models that are currently available largely rely on tumor-derived or immortalized (single) cell lines and co-cultures not representing the complex interplay between various cell types such as astrocytes, oligodendrocytes and GABA-ergic, glutamatergic, and dopaminergic neurons required for proper brain functioning [31]. Two and three-dimensional models derived from hESCs or induced pluripotent stem cells (iPSCs) consisting of various cell types can fill this gap. These cell cultures prove more complex cell models to study the dopaminergic system, the model presented in this manuscript being an example of this. Earlier steps in the characterization of this model have demonstrated that the NPCs differentiate into a heterogenic neuron-astrocyte network, developing spontaneous electrical activity within three days after differentiation and containing GABA-ergic and glutamatergic neurons [26].

In this study, this same cell culture was differentiated in a neuron-astrocyte co-culture for 70 days. Dopamine-related gene and protein expression were investigated to study the biological domain of the model. The results demonstrate that, after 42 days of differentiation, dopaminergic neurons are present. The expression of genes specific for the dopaminergic system, such as *LMX1B*, *NURR1*, *TH*, *SLC18A2*, and *KCNJ6*, was increased at day 14, and increased from day 28 onwards. Protein expression of TH, a marker for catecholaminergic neurons which are precursors of dopaminergic neurons, was expressed at day 14 and became more abundant from day 21 onwards. The VMAT2 protein, a monoamine transporter, was robustly expressed in neurons and GFAP^+^ astrocytes from day 28 onwards. While *SLC6A3* gene expression seemed downregulated, also when using different primers, DAT protein was clearly present in the cells and was increased from day 42 onwards. Overall, there was robust expression of markers that collectively indicated the presence of dopaminergic neurons in this model. While this is not the first human stem cell-based model in the field of developmental neurotoxicity research shown to contain dopaminergic neurons with TH immunostainings [32,33,34], this study was the first to additionally show the expression of transporters solely present in dopaminergic neurons, such as VMAT2 and DAT, which are pivotal for functional dopaminergic neurons.

To draw conclusions on the biological as well as toxicological applicability domain of the model described here, further characterization is necessary. This would comprise experiments to determine the size and stability of the fraction of dopaminergic neurons, including the sensitivity of the culture towards selective dopaminergic toxicants as well as specific neurotoxicants and non-neurotoxicants, e.g., using flow cytometry techniques. With these experiments, a shift in particular cell populations can be detected. Moreover, functional aspects of the model have to be assessed, including (spontaneous) electrical activity, neuronal network formation, and parameters of intracellular calcium homeostasis [35,36]. An important additional determinant of functional dopaminergic neurons is dopamine release which could be determined using electrophysiology, chemical analysis, or fluorescent probes depending on the sensitivity level required. Cell death and oxidative stress are also clearly implicated in dopaminergic system-related neurotoxicity and degeneration, and therefore the effects on cell viability, production of reactive oxygen species, and mitochondrial dysfunction are interesting endpoints to set up as well. For studying the developmental effects of compound exposure on dopaminergic neurons, measuring the dopaminergic neurons in the hNPT from day 21 to day 42 may be more relevant. Including these endpoints can increase the applicability of this more complex cell model in a testing strategy for dopamine-related neurotoxicity. The added value in comparison to other models also needs to be discussed considering the duration of the protocol (up to 70 days).

As a result of the co-culture of neurons and GFAP^+^ astrocytes, spontaneous electrical activity starts to develop within three days of differentiation [26], indicating functionality which is a clear asset as compared to single cell models with characteristics of dopaminergic neurons, such as SH-SY5Y, PC12, N27, and the LUnd Human MESencephalic (LUHMES) cell lines [22,37,38,39]. Although these models have a clear advantage for studying molecular details of pathophysiological processes, having a more complex model, the presence of astrocytes is important in the light of chemical compound effects as some compounds are metabolized into a toxic form by astrocytes, e.g., 1-methyl-4-phenyl-1,2,3,6-tetrahydropyridine (MPTP) which is metabolized into 1-methyl-4-phenylpyridinium (MPP^+^) [40]. It would be interesting to test this compound set in this in vitro model to confirm that the astrocytes present in the culture are sufficiently metabolically active to convert parent compounds into toxic metabolites.

Interestingly, the astrocytes in this model expressed high levels of VMAT2, which do not seem to be present in astrocytes in the SNpc [41], but only in the prefrontal cortex of mice and potentially humans [42,43]. This suggests that the culture might contain a population of astrocytes resembling the population the dopaminergic neurons project to, rather than the astrocytes in the SNpc. Moreover, single cell RNA-seq of human SNpc shows that the majority of the cells in this brain area are oligodendrocytes, which are not present in this model [25,44]. Rather than a mini midbrain, our model might represent the dopaminergic neurons in the midbrain and the cells they project to. This exemplifies that in vitro models need proper characterization and consideration for the intended purpose in an AOP. In addition, it also highlights the complexity of the brain and the need to set-up more complex models and place them in an AOP for neurotoxicity.

## 4. Materials and Methods

### 4.1. Growth and Differentiation of a Neuron—Astrocyte Culture That Contains Dopaminergic Neurons from Embryonic Stem Cell-Derived Neural Progenitor Cells

NPCs were differentiated from H9 human embryonic stem cells (WA09, passage 58, WiCell, Madison, WI, USA) as described before [26]. Cells were tested for mycoplasma infections annually. NPCs were thawed and cultured according to the Stemcell neuronal induction protocol (Document #28782, Stemcell Technologies, Vancouver, BC, Canada). Specifically, NPCs were maintained in STEMdiff™ Neural Progenitor (NP) medium (Stemcell) on Poly-L-Ornithine (PLO, 15 µg/mL, Sigma-Aldrich, Saint Louis, MO, USA)—laminin (10 µg/mL, Sigma-Aldrich) coated 6-well plates (Corning, New York, NY, USA) in a humidified chamber (37 °C, 5% CO_2_, 3% O_2_). The cells received daily refreshments for one week until they reached 100% confluency. To start the differentiation, NPCs were dissociated and seeded at 2.56 × 10^5^ cells/cm^2^ on either PLO-laminin-coated 12/24/48-well plates (Corning) or 8-well micro-slides (Ibidi, Gräfelfing, Germany) in NP medium, depending on the type of experiment. After one day, the medium was fully replaced for neural differentiation (ND) medium adapted from [45]. ND medium was comprised of neurobasal medium, 20 µL/mL B-27 without retinoic acid supplement (50×), 10 µL/mL N2 supplement (100×), 10 µL/mL nonessential amino acids (100×), 10 µL/mL 5000 IU/mL Penicillin/5000 µg/mL Streptomycin, 20 ng/mL recombinant glial cell line-derived neurotrophic factor (GDNF), and 20 ng/mL recombinant brain-derived neurotrophic factor (BDNF; all Gibco, Waltham, MA, USA), 200 μM ascorbic acid (Sigma-Aldrich), 1 μM dibutyryl cyclic adenosine monophosphate (Sigma-Aldrich), and 2 μg/mL laminin (Sigma-Aldrich). In the first week of differentiation, recombinant ciliary neurotrophic factor (CNTF; Gibco) was added to the ND medium at a final concentration of 10 ng/mL. Half medium refreshments were performed every 2–3 days for up to 70 days.

### 4.2. Characterisation of Dopaminergic Neurons

Dopamine-specific elements relevant for the characterization are listed in Table 1 (genetic markers) and Table 2 (protein markers). Tyrosine Hydroxylase (*TH*/TH) is a marker for catecholaminergic neurons, which is a precursor for dopaminergic neurons. There are several transcription factors that are crucial along the different stages of dopaminergic differentiation, such as LIM homeobox transcription factor 1 beta (*LMXB1*), Paired Like homeodomain 3 (*PITX3*), and Nuclear receptor subfamily 4 group A member 2 (*NR4A2*). For functional characterization, transporters in dopaminergic neurons are also included. Vesicular monoamine transporter 2 (*SLC18A2*/VMAT2) is a transporter responsible for the packaging of monoaminergic neurotransmitters such as dopamine into synaptic vesicles, and the dopamine transporter (*SLC6A3*/DAT) is responsible for the reuptake of dopamine from the synaptic cleft into the presynapse. G-protein-regulated inward-rectifier potassium channel 2 (*KCNJ6*) is implicated in excitability, neurotransmission, and modulating the effects of dopaminergic neurons.

### 4.3. RNA Isolation and qPCR

Cell samples were fixed in QIAzol (Qiagen, Hilden, Germany) and stored until further processing at −80 °C. For the timeline, five or six samples per time point were taken from two experiments (one on days 0, 1, 4, 7, 10, 14, and 28 and one on days 0, 14, 21, 28, 42, 56, 63, and 70 of differentiation). The use of “N” and “n” in the manuscript reflects the number of independent experiments and the number of biological replicates, respectively.

Whole RNA extraction was performed using the RNeasy^®^ mini kit and protocol (Qiagen), including a DNase digestion step (Qiagen). The concentration of RNA was determined using NanoDrop™1000 spectrophotometer (Nanodrop Technologies, Wilmington, DE, USA) and RNA quality was analyzed using the 2100 Bioanalyzer (Agilent Technologies, Amstelveen, The Netherlands). RNA was converted into cDNA using the high-capacity cDNA reverse transcription kit containing random hexamer primers (Applied Biosystems, Foster City, CA, USA) according to the manufacturer’s protocols. Quantification of gene expression was performed on a 7500 Fast Real-Time PCR system (Applied Biosystems) with thermal cycling conditions as follows: 95 °C for 20 s, 40 cycles of 95 °C for 3 s, 60 °C for 30 s. Primers used are listed in Table 1. The 2^−ΔΔCt^-method was employed to calculate relative gene expression [46]. Normalization was performed against an average of the housekeeping genes Hypoxanthine phosphoribosyltransferase 1 (*HPRT1*), RNA Polymerase II Subunit A (*POLR2A*), and Glucuronidase beta (*GUSB*). Statistical analysis was performed in GraphPad Prism (version 9.1.0) using a one-way ANOVA test and post-hoc Sidak’s multiple comparisons test on each gene. In the timeline experiment, two outliers for *PITX3* (days 0 and 70) were removed from the dataset.

### 4.4. Immunocytochemistry

Immunocytochemistry for dopamine-related protein expression was performed as described previously [26]. Immunostainings were performed at 14, 21, 28, 42, 56, and 70 days of differentiation. Cells were rinsed once with pre-warmed Dulbecco’s Phosphate Buffered Saline (DPBS; no calcium, no magnesium; Gibco) and fixed using pre-warmed paraformaldehyde (Electron Microscopy Sciences, Hatfield, PA, USA) at a concentration of 4% in DPBS for 30 min. Cells were washed two times for 5 min with DPBS and permeabilized with 0.2% Triton X-100 (Sigma-Aldrich) in DPBS for 5 min. After permeabilization, cells were washed again two times for 5 min, and blocked for 1 h in blocking solution (1% bovine serum albumin (BSA; *w*/*v*; Sigma-Aldrich), 0.5% Tween-20 (*v*/*v*; Sigma-Aldrich) in DPBS). Primary antibodies were applied overnight at 4 °C in 0.5% BSA/0.5% Tween-20 in DPBS (Table 2). Samples were washed two times for 5 min with DPBS and secondary antibodies were applied for 1 h in the same antibody incubation mixture (Table 2). Cells were washed two times for 5 min with DPBS, and DAPI (20 ng/mL, Sigma-Aldrich) was applied for 7 min. After a final wash, coverslips with a drop of SlowFade^®^ Diamond Antifade were mounted in each well. Imaging was performed on a Nikon A1Rsi microscope with 100× oil objective (Nikon, Tokyo, Japan) or Leica DMi8 microscope system with 20× objective (Leica, Wetzlar, Germany) using the appropriate Nikon (NIS Elements) or Leica Software (LAS X). Images were further processed in Fiji/ImageJ (version 1.53c) [47].

## Figures and Tables

**Figure 1 ijms-24-03608-f001:**
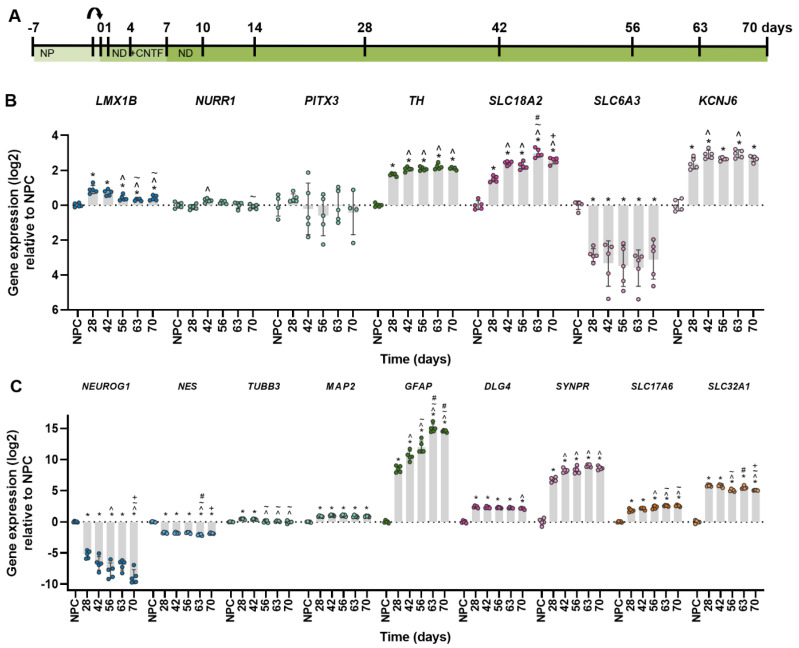
Gene expression over the course of neuronal-glial differentiation. (**A**) Timeline of the culture protocol and time points of sampling. Arrow indicates replating to a new culture plate. (**B**) Gene expression over time of a selection of dopaminergic markers for cell types relative to the neural progenitor cell (NPC) culture, ordered by expected timing of expression: early markers for differentiation of dopaminergic progenitor cells (*LMX1B*), differentiation of dopaminergic neurons (*NURR1*, *PITX3*), a catecholaminergic neuron marker (*TH*) and proteins that can transport dopamine (*SLC618A2*, *SLC6A3*) or are involved in dopaminergic neurotransmission (*KCNJ6*). (**C**) Gene expression over time of a selection of markers for cell types relative to the NPC culture: NPC (*NEUROG1*, *NES*), neurons (*TUBB3*, *MAP2*), astrocytes (*GFAP*), synapses (*SYNPR*, *DLG4*), excitatory (*SLC17A6*) and inhibitory (*SLC32A1*) neurotransmitter vesicles. NP: neural progenitor medium, ND: neural differentiation medium CNTF: Ciliary neurotrophic factor. Significance (adjusted *p*-value ≤ 0.05) is indicated relative to *: NPC, ^: 28 days, ~: 42 days, #: 56 days, +: 63 days. Full statistics can be found in Appendix A.

**Figure 2 ijms-24-03608-f002:**
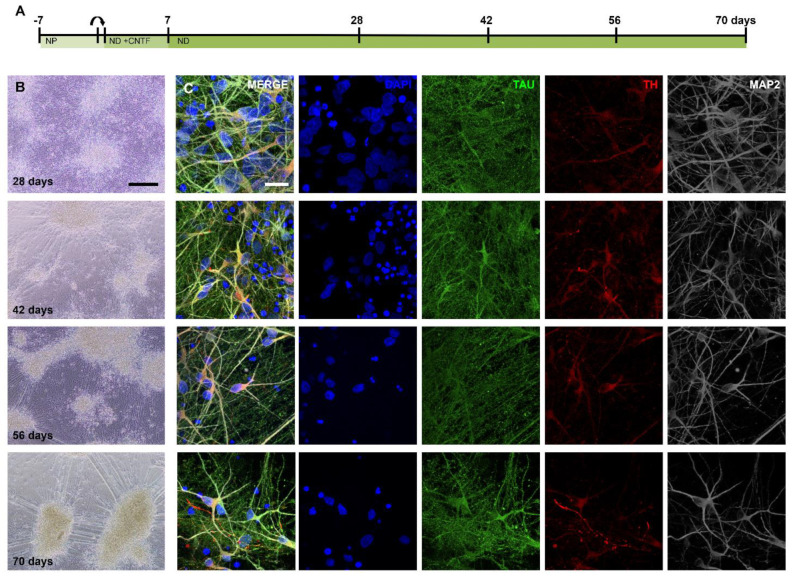
Differentiation of NPC to neuron-astrocyte culture including dopaminergic neurons. (**A**) Timeline of the culture protocol and time points of sampling. Arrow indicates replating to a new culture plate. (**B**) Light images of the neuronal culture differentiating over time. (**C**) Maximum projections (5 μm) of the same time points as in (**B**) showing neurons (TAU, MAP2) and TH^+^ neurons. Scale bars: (**B**)—200 μm, (**C**)—20 μm.

**Figure 3 ijms-24-03608-f003:**
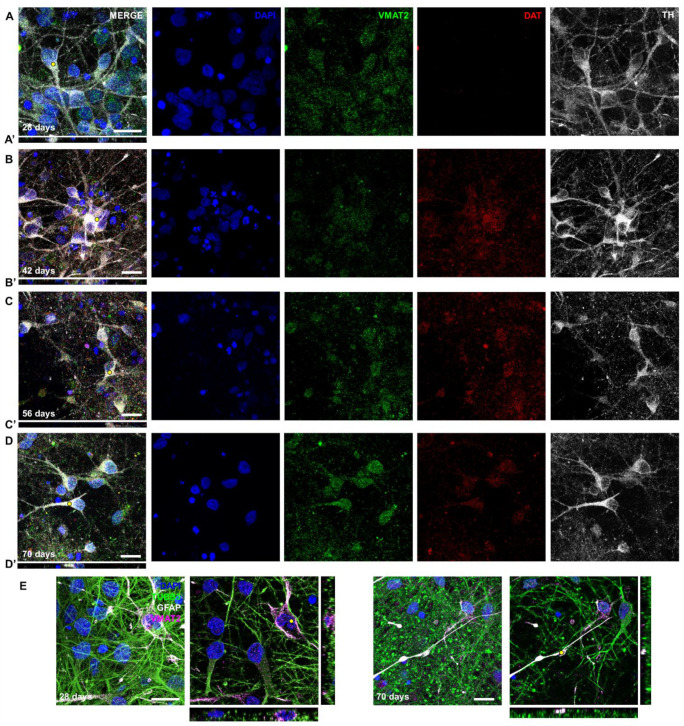
Differentiating NPCs express specific markers for dopaminergic neurons. (**A**–**D**) Maximum projections (5 μm) of the same time points as in Figure 1 showing neurons that express TH and transporter proteins VMAT2 and DAT. (**A’**–**D’**) XZ projection of (**A**–**D**). Yellow dot in corresponding image indicates where slice was made. (**E**) Maximum projections (10 μm) and single images showing intense staining of VMAT2 in GFAP^+^ astrocytes. Scale bar: 20 μm.

**Table 1 ijms-24-03608-t001:** Primers used for gene expression experiments with corresponding marker function and assay ID. All primers were purchased at Applied Biosystems. TF: transcription factor.

Gene Name	Abbreviation	Marker for	Assay ID
Neurogenin 1	*NEUROG1*	Neural ectoderm	Hs01029249_s1
Nestin	*NES*	Neural progenitor cell	Hs00707120_s1
Tubulin, beta 3 class III	*TUBB3*	Neuron	Hs00801390_s1
Microtubule-associated protein 2	*MAP2*	Mature neuron	Hs00258900_m1
Glial fibrillary acidic protein	*GFAP*	Early astrocyte	Hs00909233_m1
Synaptoporin	*SYNPR*	Pre-synapse	Hs00376149_m1
Discs Large MAGUK Scaffold Protein 4	*DLG4*	Post-synapse	Hs01555373_m1
Vesicular glutamate transporter	*SLC17A6*	Excitatory neuron	Hs00220439_m1
Vesicular GABA transporter	*SLC32A1*	Inhibitory neuron	Hs00369773_m1
LIM homeobox transcription factor 1 beta	*LMXB1*	Dopaminergic differentiation TF	Hs00158750_m1
Paired Like homeodomain 3	*PITX3*	Dopaminergic differentiation TF	Hs00374504_m1
Nuclear receptor subfamily 4 group A member 2	*NR4A2*	Dopaminergic differentiation TF	Hs00428691_m1
Tyrosine Hydroxylase	*TH*	Catecholaminergic neuron	Hs00165941_m1
Vesicular monoamine transporter 2	*VMAT2*	Monoamine transporter	Hs00996835_m1
Dopamine transporter	*SLC6A3*	Dopaminergic neuron	Hs00997374_m1
Potassium voltage-gated channel subfamily J member 6	*KCNJ6*	Dopaminergic neuron	Hs00158423_m1
Hypoxanthine phosphoribosyltransferase 1	*HPRT1*	Housekeeping gene	Hs02800695_m1
RNA polymerase II subunit A	*POLR2A*	Housekeeping gene	Hs00172187_m1
Glucuronidase beta	*GUSB*	Housekeeping gene	Hs00939627_m1

**Table 2 ijms-24-03608-t002:** Primary and secondary antibodies.

Antibody	Abbreviation	Marker for	Product Number	Company	Dilution
Rabbit anti-β-Tubulin III	TUBB3	Neuron	T2200	Sigma-Aldrich	1:1000
Guinea-pig anti Tau	TAU	Neuon, axon	314004	Synaptic Systems	1:1000
Mouse anti-Microtubule-associated protein 2	MAP2	Neuron, dendrite	801801	Biolegend	1:1000
Rabbit anti-Tyrosine hydroxylase	TH	Catecholaminergic neuron	P40101-150	Pel-Freez	1:1000
Rat anti-Glial fibrillary acidic protein	GFAP	Early astrocyte, radial glial cell	13-0300	Invitrogen	1:800
Mouse anti-Vesicular monoamine transporter 2	VMAT2	Monoamine transporter	MAB8327	R&D Systems	1:200
Rat anti-Dopamine transporter	DAT	Dopaminergic neuron	AB-N17	Advanced Targeting Systems	1:350
Goat anti-Rabbit Alexa 488			A11034	Invitrogen	1:1000
Goat anti-Guinea pig Alexa 488			A11073	Invitrogen	1:1000
Goat anti-Rabbit Alexa 555			A21429	Invitrogen	1:1000
Goat anti-Rat Alexa 555			A21434	Invitrogen	1:500
Goat anti-Mouse Alexa 647			A21236	Invitrogen	1:500

## Data Availability

The data is available in the Appendix A.

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
