# Peer review of "Prolonged Differentiation of Neuron-Astrocyte Co-Cultures Results in Emergence of Dopaminergic Neurons"

_ijms, 2023, doi:10.3390/ijms24043608_

Round 1

Reviewer 1 Report

Based on the complexity of the brain and the need to set-up more complex models in an AOP for neurotoxicity, the authors investigated  the biological domain related to dopaminergic neurons of a human stem cell based in vitro test, the human neural. According to the results, it was confirmed that there was stable gene and protein expression of dopaminergic markers in hNPT. Overall, the findings reported here are potentially interesting and constructive. However, there are some concerns that should be addressed.

Issues are listed below:

1.     Importantly, the entire manuscript should be carefully proof-read for grammatical mistakes, spacing, etc.

2.     Additionally, in the manuscript, the authors mainly observed the expression of genes specific for dopaminergic differentiation and functioning. Whether the release of neurotransmitter from dopamine neurons should also be considered.

3.     Neural progenitor cells were differentiated in a neuron-astrocyte co-culture for 70 days. In this model, some genes specific for dopaminergic differentiation and functioning were observed. The expression of specific genes can only indicate the existence of dopaminergic neurons in the differentiation system, but whether the density of dopaminergic neurons or the percentage of dopaminergic neurons should also be considered.

Author Response

We thank the three reviewers for evaluating our manuscript ID ijms-2144609 entitled " Prolonged differentiation of neuron-astrocyte co-cultures results in emergence of dopaminergic neurons” for publication in IJMS, special issue on Pharmacology of Neurodegenerative Diseases. We value the comments and the opportunity to address them and we revised the manuscript accordingly. Please find below our response to each specific comment made by the reviewers. The line numbers in this document refer to the “clean” version of the manuscript.

We hope you will find our response and modifications satisfactory. We would be much obliged with the opportunity to publish in IJMS, which we consider an excellent podium to showcase our work.

Sincerely,

Victoria de Leeuw

Reviewer comments and author responses:

Reviewer 1:

Comment to the Authors

Based on the complexity of the brain and the need to set-up more complex models in an AOP for neurotoxicity, the authors investigated the biological domain related to dopaminergic neurons of a human stem cell based in vitro test, the human neural. According to the results, it was confirmed that there was stable gene and protein expression of dopaminergic markers in hNPT. Overall, the findings reported here are potentially interesting and constructive. However, there are some concerns that should be addressed.

Issues are listed below:

  1. Importantly, the entire manuscript should be carefully proof-read for grammatical mistakes, spacing, etc.

Authors’ response

We thank the reviewer for the critical remarks. We have proof-read the manuscript and improved grammar and spelling.

  1. Additionally, in the manuscript, the authors mainly observed the expression of genes specific for dopaminergic differentiation and functioning. Whether the release of neurotransmitter from dopamine neurons should also be considered.

Authors’ response

We thank the author for this important suggestion. The goal of the current manuscript was primarily to investigate the presence of dopaminergic neurons and relevant markers in the hNPT. To further underline this model's relevance for the assessment of dopamine-specific toxicity, dopamine release should be included. However, given the scarcity of reliable methodology for measurement of dopamine release in heterogenous cultures this is considered beyond the scope of the current manuscript. As it still is an important point, we included in the discussion (line 284-288) about the additional experiments to further characterise the model, contributing to a further outline of the model's toxicological applicability domain:

“To further characterise the model, future experiments will consider the size and stability of the fraction of dopaminergic neurons as well as the sensitivity of the culture towards selective dopaminergic toxicity, using for example flow cytometry techniques. Also, functional aspects of the model will be assessed, including electrical activity and network formation, parameters of intracellular calcium homeostasis, and dopamine release.”

  1. Neural progenitor cells were differentiated in a neuron-astrocyte co-culture for 70 days. In this model, some genes specific for dopaminergic differentiation and functioning were observed. The expression of specific genes can only indicate the existence of dopaminergic neurons in the differentiation system, but whether the density of dopaminergic neurons or the percentage of dopaminergic neurons should also be considered.

Authors’ response

We agree with the reviewer's suggestion. This an issue of particular importance for heterogenic cultures. So far the results seem to demonstrate that the differentiation protocol results in a reproducible subpopulation of neurons expressing dopamine-specific markers. In future experiments we will further characterise the model using FACS-based approaches, also testing for subpopulation-directed toxicity of chemical substances. Based on the various reviewer comments, we included future experiments in the discussion where this issue is also included (see also response 2).

Reviewer 2 Report

I found that work is interesting with implications for human health with special reference to Parkinson's diseae.

1. Authors rightly observed that "this study was the first that additionally showed expression of 287 transporters that are solely present in dopaminergic neurons 

such as VMAT2 and DAT, 288 which are pivotal for functional dopaminergic neurons"

Therefore Suggest them to validate the sensitivilty of differentiated cell lines to a neurotoxicant.  

2. Can you compare the  expression of DAnergic marker such as TH with the levels in SH-SY5Y cell line.  

Author Response

We thank the three reviewers for evaluating our manuscript ID ijms-2144609 entitled " Prolonged differentiation of neuron-astrocyte co-cultures results in emergence of dopaminergic neurons” for publication in IJMS, special issue on Pharmacology of Neurodegenerative Diseases. We value the comments and the opportunity to address them and we revised the manuscript accordingly. Please find below our response to each specific comment made by the reviewers. The line numbers in this document refer to the “clean” version of the manuscript.

We hope you will find our response and modifications satisfactory. We would be much obliged with the opportunity to publish in IJMS, which we consider an excellent podium to showcase our work.

Sincerely,

Victoria de Leeuw

Reviewer comments and author responses:

Reviewer 2:

Comment to the Authors

I found that work is interesting with implications for human health with special reference to Parkinson's diseae.

  1. Authors rightly observed that "this study was the first that additionally showed expression of 287 transporters that are solely present in dopaminergic neurons 

such as VMAT2 and DAT, 288 which are pivotal for functional dopaminergic neurons"

Therefore Suggest them to validate the sensitivilty of differentiated cell lines to a neurotoxicant.  

Authors’ response

We agree with the reviewer that testing the sensitivity of this heterogenic culture to subpopulation-specific neurotoxicants is an important next step. We have added this suggestion to the paragraph on future experiments.

  1. Can you compare the  expression of DAnergic marker such as TH with the levels in SH-SY5Y cell line. 

Authors’ response

While we understand the comment of the reviewer, the short answer to this question is no. This would imply the comparison of a single marker between a homogenic and a heterogenic cell model. The SH-SY5Y cell line is considered a relevant cell model to use as a screening assay on how compounds might affect dopaminergic neurons. This cell line is simpler than the hNPT since it only contains one type of neurons and no astrocytes, and therefore, no neuron-astrocyte co-culture is formed. The hNPT can be used to test more complex outcomes and interplays of a neuronal network on the functioning of dopaminergic neurons. Therefore, this cell model will have additional value compared to the SH-SY5Y model. As TH does not provide a marker specific for dopaminergic neurons (also noradrenergic neurons stain positive for TH), we do not consider it relevant to compare the levels of TH between the cell systems.

Reviewer 3 Report

In this article, the authors present a professionally and conscientiously performed methodological study on the development of an original model of neuronal-astrocytic coculture for research and testing in vitro of the mechanisms of action of toxins on dopaminergic neurons of the human brain.

The results of the study are carefully analyzed and illustrated in detail. This model convincingly demonstrates stable expression of markers that collectively indicate the presence of dopaminergic neurons. The authors determined the optimal time interval in the propagation differentiation of model co-culture for testing dopamine-related neurotoxicity.

The model is very relevant. The results are convincing in the prospect of both direct use of the new neuron-astrocyte co-culture and its wider application to study the cellular and network neuro-glial mechanisms of the pathogenesis and development of dopamine-dependent human brain disorders.

At the same time, the authors, understanding the complexity of the  brain organization, express the conviction that their model puts not a dot, but a comma in modeling of mediator-specific brain cultures.

Below I place two small remarks.

(1)In the Introduction, you should briefly highlight the state of the art in science on the issue of modeling cultures containing dopamine neurons from the human brain and specify your task on this basis.

(2)In supplementary material, in the comments to both Tables, it should be corrected P value to P < 0.0001 for 4* significance.

Author Response

We thank the three reviewers for evaluating our manuscript ID ijms-2144609 entitled " Prolonged differentiation of neuron-astrocyte co-cultures results in emergence of dopaminergic neurons” for publication in IJMS, special issue on Pharmacology of Neurodegenerative Diseases. We value the comments and the opportunity to address them and we revised the manuscript accordingly. Please find below our response to each specific comment made by the reviewers. The line numbers in this document refer to the “clean” version of the manuscript.

We hope you will find our response and modifications satisfactory. We would be much obliged with the opportunity to publish in IJMS, which we consider an excellent podium to showcase our work.

Sincerely,

Victoria de Leeuw

Reviewer comments and author responses:

Reviewer 3:

Comment to the Authors

In this article, the authors present a professionally and conscientiously performed methodological study on the development of an original model of neuronal-astrocytic coculture for research and testing in vitro of the mechanisms of action of toxins on dopaminergic neurons of the human brain.

The results of the study are carefully analyzed and illustrated in detail. This model convincingly demonstrates stable expression of markers that collectively indicate the presence of dopaminergic neurons. The authors determined the optimal time interval in the propagation differentiation of model co-culture for testing dopamine-related neurotoxicity.

The model is very relevant. The results are convincing in the prospect of both direct use of the new neuron-astrocyte co-culture and its wider application to study the cellular and network neuro-glial mechanisms of the pathogenesis and development of dopamine-dependent human brain disorders.

At the same time, the authors, understanding the complexity of the  brain organization, express the conviction that their model puts not a dot, but a comma in modeling of mediator-specific brain cultures.

Below I place two small remarks.

  1. In the Introduction, you should briefly highlight the state of the art in science on the issue of modeling cultures containing dopamine neurons from the human brain and specify your task on this basis.

Authors’ response

We thank the reviewer for the compliments. Regarding this first point, we have added the following in the introduction to highlight the current issues and our task in this (line 69-73):

There is great progression in differentiation of human stem cells towards a dopaminergic fate for therapeutical purposes (e.g. Hiller et al., 2022; Kim et al. 2022), however, cell cultures, especially co-cultures, suited for the assessment of chemical exposure specifically to dopaminergic neurons are not yet well-developed, but urgently needed (Heusinkveld and Westerink, 2017; Meerman et al., 2022; Tebby et al., 2022).

  1. In supplementary material, in the comments to both Tables, it should be corrected P value to P < 0.0001 for 4* significance.

Authors’ response

The adjusted p value classification for **** was changed to 0.0001.

Round 2

Reviewer 1 Report

Due to the influence of some objective factors, some indicators have not yet been carried out, so it is suggested to further improve the discussion.

Author Response

We thank the three reviewers for the quick response and we revised the manuscript accordingly. Please find below our response to each specific comment made by the reviewers. The line numbers in this document refer to the “clean” version of the manuscript.

We hope you will find our response and modifications satisfactory. We would be much obliged with the opportunity to publish in IJMS, which we consider an excellent podium to showcase our work.

Sincerely,

Victoria de Leeuw

Reviewer comments and author responses:

Reviewer 1:

Comment to the Authors

Due to the influence of some objective factors, some indicators have not yet been carried out, so it is suggested to further improve the discussion.

Authors’ response

We agree that we may have not elaborated on some of the points mentioned by the reviewer. We have edited the discussion paragraph that was inserted in the previous review round (line 284-293):

“To draw conclusions on the biological as well as toxicological applicability domain of the model described here, further characterization is necessary. This would comprise experiments to determine the size and stability of the fraction of dopaminergic neurons including the sensitivity of the culture towards selective dopaminergic toxicants as well as a-specific neurotoxicants, and non-neurotoxicants using for example flow cytometry techniques. With these experiments a shift in particular cell populations can be detected. Also, functional aspects of the model have to be assessed, including (spontaneous) electrical activity, neuronal network formation, and parameters of intracellular calcium homeostasis (Gleichmann and Mattson, 2011; Surmeier et al. 2011). An important additional determinant of functional dopaminergic neurons is dopamine release which could be determined using electrophysiology, chemical analysis or fluorescent probes depending on the sensitivity level required.”

Reviewer 2 Report

  1. Authors rightly observed that "this study was the first that additionally showed expression of 287 transporters that are solely present in dopaminergic neurons

such as VMAT2 and DAT, 288 which are pivotal for functional dopaminergic neurons"

Therefore Suggest them to validate the sensitivilty of differentiated cell lines to a neurotoxicant.

Authors’ response

We agree with the reviewer that testing the sensitivity of this heterogenic culture to subpopulation-specific neurotoxicants is an important next step. We have added this suggestion to the paragraph on future experiments

Rev Comment:

Glad that this aspect is added up as- future experiments.  However I suggest the authors to include data relating to just one (single) concentration of the neurotoxicant studies.  Detailed study can be included in the subsequent manuscript.  This addition will not affect the publication chances of the second manuscript.  

Author Response

Reviewer comments and author responses:

Reviewer 2:

Comment to the Authors

Authors rightly observed that "this study was the first that additionally showed expression of 287 transporters that are solely present in dopaminergic neurons

such as VMAT2 and DAT, 288 which are pivotal for functional dopaminergic neurons"

Therefore Suggest them to validate the sensitivilty of differentiated cell lines to a neurotoxicant.

Authors’ response

We agree with the reviewer that testing the sensitivity of this heterogenic culture to subpopulation-specific neurotoxicants is an important next step. We have added this suggestion to the paragraph on future experiments

Rev Comment:

Glad that this aspect is added up as- future experiments.  However I suggest the authors to include data relating to just one (single) concentration of the neurotoxicant studies.  Detailed study can be included in the subsequent manuscript.  This addition will not affect the publication chances of the second manuscript.

Authors’ response

We appreciate the suggestion of the reviewer and once again agree on this important second tier characterisation of the model. However, in order to do a proper toxicological study on the specific sensitivity of dopaminergic cells in a heterogenic culture to selective (non-)neurotoxicants, one would need to test full dose-response curves of compounds specifically toxic to neurons, general neurotoxicants and non-neurotoxicants, and quantify changes in cell populations using flow cytometry or imaging techniques. To do this in a scientifically sound manner, this would comprise an entire new study. With all due respect, inclusion of one (single) concentration of the neurotoxicant cytotoxicity studies would therefore not be of added value to the current manuscript as the current piece of work provides a first tier characterisation of the model.
